# Experiences of Caregivers and At-Risk Children Enrolled in a Prospective Pregnancy-Birth Cohort Study into the Causes of Type 1 Diabetes: The ENDIA Study

**DOI:** 10.3390/children10040637

**Published:** 2023-03-29

**Authors:** Kelly J. McGorm, James D. Brown, Alison G. Roberts, Susan Greenbank, Daniella Brasacchio, Alyssa C. P. Sawyer, Helena Oakey, Peter G. Colman, Maria E. Craig, Elizabeth A. Davis, Georgia Soldatos, Rebecca L. Thomson, John M. Wentworth, Jennifer J. Couper, Megan A. S. Penno

**Affiliations:** 1Discipline of Paediatrics, Adelaide Medical School, Robinson Research Institute, The University of Adelaide, Adelaide, SA 5005, Australia; 2Rio Tinto Children’s Diabetes Centre, Telethon Kids Institute, University of Western Australia, Perth, WA 6009, Australia; 3Department of Endocrinology and Diabetes, Perth Children’s Hospital, Perth, WA 6009, Australia; 4Community and Consumer Involvement Group (CCIG), ENDIA Study, Adelaide, SA 5005, Australia; 5School of Psychology, The University of Adelaide, Adelaide, SA 5005, Australia; 6Diabetes and Endocrinology, Royal Melbourne Hospital, Melbourne, VIC 3050, Australia; 7The School of Women’s and Children’s Health, Faculty of Medicine, University of New South Wales, Sydney, NSW 2052, Australia; 8Institute of Endocrinology and Diabetes, The Children’s Hospital at Westmead, Sydney, NSW 2145, Australia; 9Monash Centre for Health Research and Implementation, School of Public Health and Preventive Medicine, Monash University, Melbourne, VIC 3004, Australia; 10Diabetes and Vascular Medicine Unit, Monash Health, Melbourne, VIC 3175, Australia; 11Walter and Eliza Hall Institute of Medical Research, Melbourne, VIC 3052, Australia; 12Diabetes and Endocrinology, Women’s and Children’s Hospital, North Adelaide, SA 5006, Australia

**Keywords:** type 1 diabetes, cohort study, evaluation, consumer and community involvement, consumer engagement

## Abstract

Background: We sought research experiences of caregivers and their children were enrolled in the Environmental Determinants of Islet Autoimmunity (ENDIA) study. Methods: ENDIA is a pregnancy–birth cohort investigating early-life causes of type 1 diabetes (T1D). Surveys were sent to 1090 families between June 2021 and March 2022 with a median participation of >5 years. Caregivers completed a 12-item survey. Children ≥ 3 years completed a four-item survey. Results: The surveys were completed by 550/1090 families (50.5%) and 324/847 children (38.3%). The research experience was rated as either “excellent” or “good” by 95% of caregivers, and 81% of children were either “ok”, “happy” or “very happy”. The caregivers were motivated by contributing to research and monitoring their children for T1D. Relationships with the research staff influenced the experience. The children most liked virtual reality headsets, toys, and “helping”. Blood tests were least liked by the children and were the foremost reason that 23.4% of the caregivers considered withdrawing. The children valued gifts more than their caregivers. Only 5.9% of responses indicated dissatisfaction with some aspects of the protocol. The self-collection of samples in regional areas, or during the COVID-19 pandemic restrictions, were accepted. Conclusions: This evaluation identified modifiable protocol elements and was conducted to further improve satisfaction. What was important to the children was distinct from their caregivers.

## 1. Introduction

Participant retention in longitudinal cohort studies is challenging. When children are active participants, it is important to understand their experiences, as well as those of their caregivers, in the design, planning, and implementation of research. When interviewed about their involvement in clinical research, children reported that they felt positive about it and they like “helping” [1,2,3]. Children seek an active part in decision making [4,5] but they may show limited, age-appropriate understanding of research [6]. There are ethical and financial obligations to ensure that the benefits of research justify the possible negative experiences of adults and children, and to maximize their long-term participation [7,8]. Understanding the perceptions of children is increasingly relevant in type 1 diabetes (T1D), where the research initiatives for preclinical risk testing have commenced in some populations, including among those without a family history of T1D [9,10].

Here, we report the findings of an evaluation survey that was completed by participants in the Environmental Determinants of Islet Autoimmunity (ENDIA) Study. ENDIA is an Australia-wide pregnancy–birth cohort study of children who are at risk of T1D on account of having a first-degree relative (FDR) with T1D. ENDIA aims to identify the prenatal and the early-life environmental exposures that drive the development of islet autoimmunity, leading to T1D. The ENDIA protocol was published recently [11]. Children are required to provide biological samples every 3 months for the first 2 years of life, and 6 monthly thereafter. The study also requires their caregivers to provide some biological samples and complete questionnaires around their nutrition and lifestyle. Families living more remotely participate via the regional program where caregivers collect biological samples themselves and return them via courier to the lab [12]. A follow-up of the cohort continues until children turn 10 years of age. Comparing mothers with and without diabetes has enabled the study team to investigate the impact of maternal T1D on the virome [13], microbiome [14] and mycobiome [15]. Papers on the primary outcome, the development of islet autoimmunity and type 1 diabetes in children, are anticipated in 2024.

The objective of the ENDIA Participant Experience Survey (EPES) was to evaluate the experiences of caregivers and children aged ≥3 years in the following areas: (i) whether their expectations of ENDIA were being met, (ii) their motives for enrolling and remaining in the study, (iii) their reasons for leaving the study, (iv) their satisfaction with the study protocols, including the Regional Participation Program (RPP) and the protocol changes made in response to COVID-19 restrictions [16], and (v) their overall experience of being involved.

## 2. Materials and Methods

### 2.1. Participants

ENDIA recruitment occurred from 2013 to 2019 [17], resulting in a study population of 1214 unique gestational mothers, 1217 unique genetic fathers, and 1473 babies. In June 2021, 1090/1214 mothers (89.8%) were still participating and their family units were invited to complete the EPES (Figure 1A), which represents 1336 children of whom 846 were aged ≥3 years (Figure 1B). 

### 2.2. Survey Design

The EPES comprised two sections, targeting caregivers (Caregiver Survey) and children (Child Survey; Online Appendix A). 

#### 2.2.1. Caregiver Survey

The Caregiver Survey was adapted from The Environmental Determinants of Diabetes in the Young (TEDDY) Study Parent Experiences Questionnaire [18], with permission from the TEDDY Psychosocial Committee. The survey covered 12 items that explored study elements important to participants, their overall experience and satisfaction, their suggested improvements, and their thoughts on leaving the study. The responses were rated on a Likert scale with negative, neutral, and positive options, and caregivers could provide free-text comments to all questions. Additional questions were specific to the RPP and the impact of the COVID-19 pandemic on participation. 

#### 2.2.2. Child Survey

The first item of the Child Survey, developed by the ENDIA Consumer and Community Involvement Group (CCIG), and comprising volunteering ENDIA caregivers and other advocates from the T1D community, measured how the children felt about being in the study based on a modified Faces Rating Scale [19] (range: “very happy” to “very unhappy”). The children were also asked what they liked and disliked about ENDIA, and if anyone assisted them in completing the survey.

### 2.3. Survey Collection

Hardcopies of the EPES were mailed to 1090 families enrolled in March 2021. Simultaneously, hyperlinks to an electronic version of the survey were emailed to all caregivers who had an address on file. In the electronic version, the Child Survey was triggered only after the Caregiver Survey was completed; thus, the Child Survey could not be completed electronically in isolation. For families with multiple children enrolled in ENDIA, the Caregiver Survey was only completed once but all participating children aged ≥3 years were invited to complete the Child Survey. The surveys were open for a nine-month period from June 2021 to March 2022, which gave all families the opportunity to attend at least one study visit during the survey period. Email reminders were sent in July, August, and September 2021. A final reminder with a survey link was sent via mobile phone text message to those who had not responded in March 2022. Responses were collected, entered, and managed using REDCap [20,21].

### 2.4. Data Analysis

#### 2.4.1. Quantitative Data

EPES data were imported to R version 4.1.0 [22] and linked to ENDIA’s demographic information. Data not normally distributed were log transformed prior to analysis. Chi-squared tests of independence for categorical variables and two samples of Welch’s t-tests for continuous variables were used to determine differences between: (i) completers and non-completers, (ii) those who did and did not have thoughts of leaving, and (iii) those who rated the study as “good” or “excellent” versus “terrible”, “bad”, or “average”. Transformations were used to satisfy model assumptions for variables with non-normal error distributions. Poisson Generalized Linear Models were used for the count variables. The Spearman’s rank correlation test in the pspearman package [23] was used to compare the overall ratings by children and their caregivers. The significance level for all tests was 5%. Descriptive statistics were displayed using a combination of tables (such as the kableExtra package [24]) and plots (such as the ggplot2 [25]), and Likert packages [26] with Likert plots shown for questions with ordinal response scales.

#### 2.4.2. Qualitative Data

Free-text comments were categorized using a content analysis approach and analyzed separately by two researchers (AR and KM). Examples of comments were provided in Online Appendix A. The categorized responses were displayed as bar plots showing the frequencies of each category. Free-text comments from the Child Survey were entered into the tidytext R package that implements text-mining functions. After removing common “stop” words (e.g., “the”), the most frequent likes and dislikes were displayed graphically using ggplot2 [25].

## 3. Results

### 3.1. Response Rate

The Caregiver Survey achieved a 50.5% response rate (550/1090; Figure 1A). Three caregivers completed the survey anonymously; thus, 547 responses were linked to demographic data. Of the children aged ≥3, 324/847 completed the Child Survey (38.3%; Figure 1B). Of the completed surveys, 234/550 Caregiver Surveys (42.5%) and 217/324 Child Surveys (67.0%) were completed in hardcopy. Of the families who responded, 62% had attended a study visit within 6 months prior to completing the survey.

### 3.2. Demographics

Caregivers who identified as mothers (96%) completed most of the surveys, while only 25 fathers responded (Table 1). Eight caregivers (1.2%) did not specify their relationship with the ENDIA child. When compared with non-completer families, completers were more likely to have a paternal T1D proband, have attended a study visit within the last six months, and live in New South Wales and Victoria. In completer families, the mean maternal age was significantly higher and the mean age of their ENDIA child(ren) was lower despite the equivalent durations of study participation (Table 1). There were no significant differences between completers and non-completers according to the time of enrolment in the ENDIA study (prenatally or postnatally), the number of children enrolled, or whether the ENDIA child was positive for islet or coeliac autoantibodies (Table 1).

### 3.3. Overall Caregiver and Child Experience

The overall participant experience of ENDIA was positive, with 95% of caregivers indicating that it was “excellent” (63%) or “good” (32%) (Figure 2A). Those who rated their study experience as “terrible”, “bad”, or “average” were more likely to be inactive and were predominantly from one study site. Of the 324 child completers, 28.4% felt “ok”, 26.9% “happy”, and 25.6% “very happy” to be involved in ENDIA (Figure 2B). The matched caregiver–child responses for 292 dyads revealed a positive correlation between the caregiver’s and the child’s overall rating with a Spearman’s rank correlation of 0.33 (*p* < 0.001; Figure 2C). The relationships between study staff, caregivers, and children were commonly associated with positive experience ratings. Caregivers reported that they valued their child’s interaction with the study staff more highly than their own interactions with staff, with 84% versus 67% considering this to be “very important”, respectively (Figure 2D).

### 3.4. Expectations of Study Participation

Regarding caregivers, 82.3% reported that the information provided at the time of enrolment “mostly” or “completely” prepared them for participation in ENDIA, while only 2.3% reported they were not prepared. A quarter of participants (24.8%) felt “surprised” by elements of ENDIA; the appreciation from staff was a positive factor, while the food diary completion, and the frequency, volume and duration of visits were negative surprises (Figure 3A).

### 3.5. Motives for Participating in ENDIA

Caregivers reported that helping to discover the causes of T1D, preventing T1D, and monitoring their child for T1D were the most important reasons for staying in ENDIA (Figure 3B). The children most liked the virtual reality (VR) headsets used as a distraction during blood draws (Figure 3C). However, the blood draws themselves, also described as “ouches”, “hurts” or “needles”, were the most disliked aspects by the children. The children valued the gifts/small toys (Figure 3C), which caregivers considered one of the least important aspects of participation (Figure 2D). The children also commonly mentioned they liked the nice staff and “helping” (Figure 3C). 

### 3.6. Thoughts of Leaving

Most completers (76.6%) had never thought of leaving ENDIA. Those who had were more likely to have been in the study for a longer time, had older children, were more likely to be inactive, and tended to participate at a particular study site (Table 2).

The commonly cited reasons for leaving the study included blood tests, the time commitment, food diaries and diet-recall surveys (Figure 4A).

### 3.7. Satisfaction with the Study Protocol

The caregivers indicated the highest levels of satisfaction with the collection of swabs, urine and stool samples, and the receipt of vouchers for the out-of-pocket expenses incurred to attend study visits (Figure 4B). The lower levels of satisfaction were related to the ENDIA study app and diet-recall questionnaires, although participants still reported being predominantly satisfied or very satisfied with these activities. The average dissatisfaction rating across the eight areas of the study that were surveyed was only 5.9%.

### 3.8. Regional Participation Program (RPP) 

The participants in the RPP collected their own child’s samples and attended regional pathology centers for venipuncture [12]. Despite this extra work, the 66 regional completers who responded to the question were generally satisfied, although 20% reported dissatisfaction with the external courier (Figure 4C).

### 3.9. ENDIA during COVID-19

Participants supported the continuation of ENDIA throughout the COVID-19 pandemic restrictions enforced in Australia from March 2020 to July 2021, which necessitated some changes to the protocol [16]. The lack of face-to-face contact and the reduced frequency of blood tests during lockdowns had the greatest impact on participation (Figure 4D). The self-collection and storage of samples in home freezers was generally not considered problematic.

## 4. Discussion

We reported the experiences of 550 caregivers, predominantly mothers, and 324 children aged ≥3 years, participating in a large pregnancy–birth cohort of children at risk of developing T1D. This work was unique in four major ways: (1) ENDIA was the first study in the world to establish a cohort at risk of developing type 1 diabetes from pregnancy; (2) we included the responses of a substantial number of young children; (3) we reported the perceptions of families participating in regional areas of Australia; and (4) the timing of the evaluation occurred during the global COVID-19 pandemic, providing a unique insight into the challenges of participating in a longitudinal study and collecting biological samples at this time.

Most respondents reported positive overall experiences. For caregivers, the relationship and communication with research nurses, the contribution to T1D research, and the ability to monitor their child(ren) for early signs of T1D were the key reasons for positive experiences and the decision to stay in the study. The prompt and sensitive reporting of results was highly valued. 

The completer mothers tended to be older and their children were younger. The clinical relevance of these points of difference is unclear. Assumptions about older mothers with younger children could be that they have more life experience; perhaps delaying childbirth for education and career purposes. Future cohort studies are encouraged to look at ways of targeting younger parents to engage them, such as considering the use of more modern tools like Snapchat and TikTok. However, the ethical review and the use of these tools could prove challenging in the research context. 

The child participants in ENDIA were most engaged by VR headsets and the receipt of thank-you gifts or toys. The VR headsets were implemented in 2019 following reports of their benefits [27]. Almost 40% of ENDIA children indicated that VR headsets were the aspect of the study they liked the most. The blood draw, for which the VR headset was used, was the most disliked aspect, and the major reason caregivers considered leaving the study (Figure 3). This juxtaposition indicates that the children were disassociating the VR experience from the venipuncture, which was the intent. The relative importance of thank-you gifts was a point of difference between caregivers and children. Almost half of caregivers (48%) ranked them as “unimportant”. 

Our findings were similar, in part, to those reported by the TEDDY study [18], which followed children who also had increased genetic risk of T1D in the USA, Finland, Germany, and Sweden for the first months of life. TEDDY’s caregiver evaluation survey response rate (59.1%) was comparable; most respondents were mothers and the majority were also satisfied with their study experience. These comparable results indicate that cohort studies demand a great deal of commitment from research participants. Researchers need to be very clear about the value, and careful about the load, that additional surveys place on their participants. Administering the survey where the cohort was around a median of 4 years meant that those who were dissatisfied or unable to continue participation had likely already withdrawn. TEDDY indicated that withdrawal was highest in the first year [28]. Those remaining in ENDIA were possibly satisfied enough with the study to continue and not feel the need to evaluate it.

EPES completer families were more likely to have fathers with T1D. TEDDY found that unsupportive partners, who were disinterested in the research, made engagement and follow-up difficult for the primary caregiver [29].

The reasons for positive or negative caregiver experiences in TEDDY were similar to those of ENDIA caregivers, suggesting there are similar attitudes towards research participation across the USA, Europe and Australia. A similar proportion of TEDDY caregivers had thoughts of leaving (24% for TEDDY and 23% for ENDIA) citing the blood draws, time, protocol demands, and food diaries as reasons. High staff turnover negatively affected participant satisfaction in TEDDY [30]. We also found differences in EPES completion and satisfaction between the study sites. Sites with more stable staffing tended to report higher satisfaction. This further emphasizes the value of retention and support for frontline research staff in longitudinal cohorts. The experiences of young children were not reported in TEDDY; thus, the distinction between the caregiver’s and the child’s experiences reported herein is novel in T1D research. 

### 4.1. Limitations

Our study had some limitations. The response rate reached was 50.5% despite efforts to engage participants via multiple points of contact. This indicates the burden of additional requests on this cohort, a reluctance of some caregivers to involve their children in further surveys, and perhaps a perception that their children would not understand the concepts being asked of them. The child survey relied on the primary caregiver passing on the questionnaire to the child, as well as assisting them with completion, due to their young age (median 5.6 years). Parental assistance potentially influenced children’s responses. 

The impact of the COVID-19 pandemic on response rates may also play a part. During the survey period, caregivers balanced study participation, work, school, COVID-19 restrictions and other obligations. The increased load on caregivers around the peak time of COVID-19 lockdowns etc. has been reported previously [31]. 

Responders may have been more likely to report satisfaction, thus introducing reporting bias. In line with previous research, children may have been more likely to rate their satisfaction as lower [32]. 

### 4.2. Implications

To ensure families feel informed, we provide the ENDIA study results and publications through multiple media, including direct email, social media [16], newsletters, and summary updates for staff to provide at visits. We are developing age-appropriate resources to further explain study involvement to children, especially sample collections such as blood tests and stool, in consultation with our CCIG. 

The introduction of VR headsets to assist children with blood taking, and other uncomfortable study procedures, proved to be a worthwhile investment. Because the children placed a high value on gifts, we strongly encourage other researchers to provide child-focused and age-appropriate acknowledgements of their participation. Some caregivers even cited their discontent with the toys in the free-text responses as unnecessary “plastic” (refer to Online Appendix A). In response to this, we are investigating more sustainable options. 

We plan to administer the child survey again when the children turn 10 years old, which is at study exit, to compare these findings.

## 5. Conclusions

Our work provides several lessons for conducting early-life observational studies. A trusted relationship with the research staff greatly influenced the participation experience. The caregivers highly valued flexibility, appreciation, continuity, and psychosocial support, highlighting that frontline staff retention is a critical component of study-participant retention. The opportunity to advance research and the ability to monitor their children’s risks were also very important to caregivers, and highlight the need for the ongoing results of the study to be disseminated to participants. We also demonstrated that eliciting the child’s voice in research experience is feasible and worthwhile, and their experiences may be different to those of their caregivers.

## Figures and Tables

**Figure 1 children-10-00637-f001:**
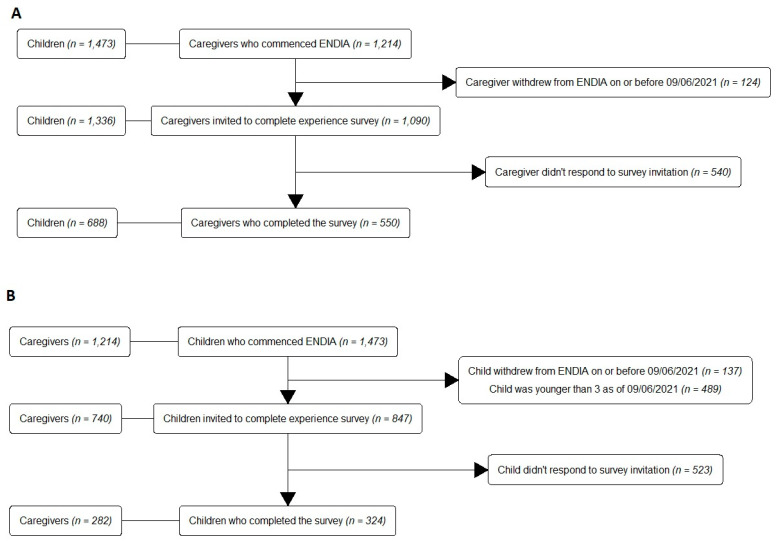
STROBE diagrams for (**A**) the Caregiver Survey, and (**B**) the Child Survey.

**Figure 2 children-10-00637-f002:**
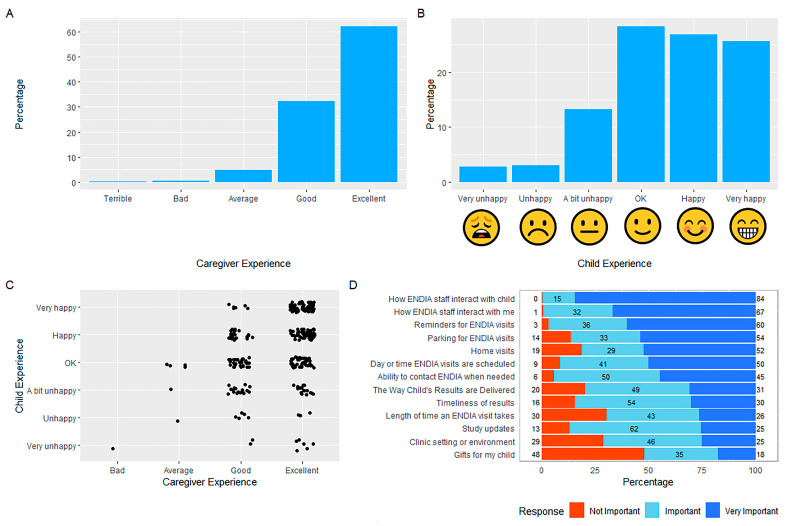
Overall experience ratings for (**A**) the Caregiver Survey, and (**B**) the Child Survey. (**C**) Correlated caregiver-child responses for overall experience ratings. (**D**) Responses to “How important have the following parts of the study been for your experience in ENDIA?”

**Figure 3 children-10-00637-f003:**
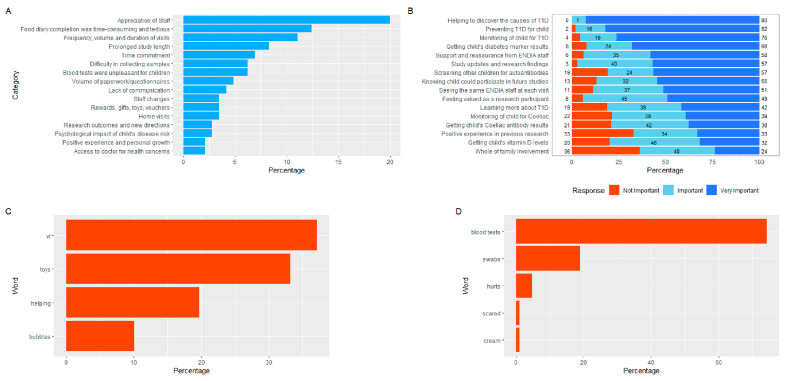
(**A**) Aspects of participation in ENDIA that caregivers found to be “surprising”, which reflect both positive and negative experiences. (**B**) Responses to “How important have the following reasons been for staying in ENDIA?” (**C**) Commonly reported “likes” from the Child Survey. (**D**) Commonly reported “dislikes” from the Child Survey.

**Figure 4 children-10-00637-f004:**
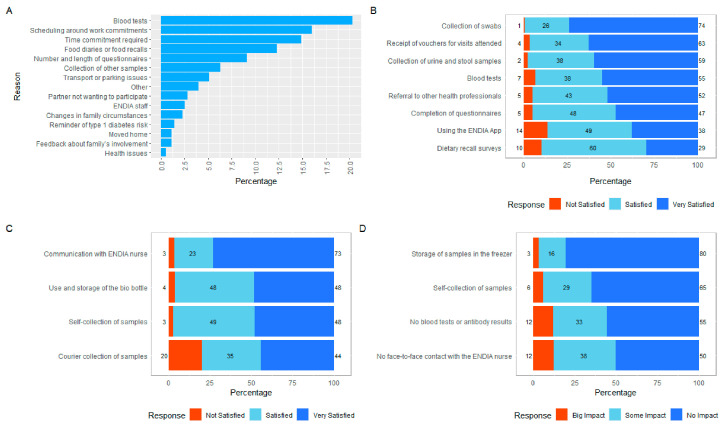
(**A**) Reasons provided by caregivers for thoughts of leaving ENDIA. (**B**) Caregiver satisfaction with the study protocol. (**C**) Caregiver satisfaction with the RPP. (**D**) Impact on the caregiver of the protocol changes implemented during COVID-19 restrictions.

**Table 1 children-10-00637-t001:** Characteristics of ENDIA families (1.a: all parents, 1.b: all children) who did and did not complete the EPES.

**1.a. Parental Characteristics**	**Completers (n = 547)**	**Non-Completers (n = 543)**	***p*-Value**
*Age of Mother (years)*			
Mean (SD)	37.8 (4.87)	37.1 (5.10)	0.016
*Length of Study (years)*			
Median [Q1, Q3]	5.13 [4.02, 6.41]	5.37 [4.19, 6.42]	0.073 ^a^
*Age of Father (years)*			
Mean (SD)	39.6 (5.75)	39.3 (5.81)	0.45
Missing	21 (3.8%)	32 (5.9%)	
*Number of Children in ENDIA*			
One	421 (76.97%)	440 (81.03%)	0.106
Two	114 (20.84%)	98 (18.05%)	
Three	12 (2.19%)	5 (0.92%)	
**1.b. Child Characteristics**	**Completers (n = 685)**	**Non- Completers (n = 651)**	***p*-Value**
*Age of Child (years)*			
Mean (SD)	4.93 (1.73)	5.17 (1.75)	0.012
Missing	0 (0%)	1 (0.2%)	
*Days Since Last Visit*			
Median [Q1, Q3]	127 [65.5, 194]	229 [91.0, 805]	<0.001 ^b^
Missing	7 (1.0%)	8 (1.2%)	
*Current ENDIA Participation*			
Active	648 (94.60%)	466 (71.58%)	<0.001
Inactive	37 (5.40%)	185 (28.42%)	
*Proband Relationship(s)*			
Mother only	395 (57.66%)	413 (63.44%)	0.02
Father only	209 (30.51%)	154 (23.66%)	
Sibling only	61 (8.91%)	70 (10.75%)	
Multiple FDR probands ^c^	19 (2.77%)	11 (1.69%)	
Other ^d^	1 (0.15%)	3 (0.46%)	
*Location*			
Site A	202 (29.49%)	159 (24.42%)	<0.001
Site B	158 (23.07%)	123 (18.89%)	
Site C	78 (11.39%)	116 (17.82%)	
Site D	93 (13.58%)	103 (15.82%)	
Site E	66 (9.64%)	93 (14.29%)	
Regional	88 (12.85%)	57 (8.76%)	
*Recruitment Time*			
Prenatal	561 (81.90%)	522 (80.18%)	0.466
Postnatal	124 (18.10%)	129 (19.82%)	
*Islet autoantibodies Positive*			
Yes	40 (5.84%)	38 (5.84%)	1
No	645 (94.16%)	613 (94.16%)	
*Coeliac autoantibodies Positive*			
Yes	8 (1.17%)	14 (2.15%)	0.232
No	677 (98.83%)	637 (97.85%)	

^a^ Log transformed; ^b^ Poisson regression; ^c^ Multiple first-degree relatives with T1D; ^d^ Includes donor gametes with genetic first degree relative (FDR) with T1D.

**Table 2 children-10-00637-t002:** Demographics of caregivers with thoughts of leaving ENDIA based on 524/550 responses to the question.

Thoughts of Leaving	No (*n* = 402)	Yes (*n* = 122)	*p*-Value
*Length of Study (years)*			
Median [Q1, Q3]	4.64 [3.83, 6.04]	5.41 [4.19, 6.57]	0.003 ^a^
*Age of Child (years)*			
Mean (SD)	4.82 (1.67)	5.39 (1.87)	0.003
*Days Since Last Visit*			
Median [Q1, Q3]	125 [62.0, 183]	152 [76.5, 320]	0.001 ^b^
Missing	3 (0.7%)	3 (2.5%)	
*Current ENDIA Participation*			
Active	387 (78.18%)	108 (21.82%)	0.002
Inactive	15 (51.72%)	14 (48.28%)	
*Proband Relationship(s)*			
Mother only	235 (76.30%)	73 (23.70%)	0.78
Father only	120 (78.95%)	32 (21.05%)	
Sibling only	37 (75.51%)	12 (24.49%)	
Multiple FDR probands ^c^	9 (64.29%)	5 (35.71%)	
Other ^d^	1 (100.00%)	0 (0.00%)	
*Location*			
Site A	129 (82.69%)	27 17.31%)	<0.001
Site B	90 (78.26%)	25 (21.74%)	
Site C	32 (49.23%)	33 (50.77%)	
Site D	66 (94.29%)	4 (5.71%)	
Site E	51 (73.91%)	18 (26.09%)	
Regional	34 (69.39%)	15 (30.61%)	
*Recruitment Type*			
Prenatal	333 (77.44%)	97 (22.56%)	0.481
Postnatal	69 (73.40%)	25 (26.60%)	
*IA Positive*			
Yes	21 (77.78%)	6 (22.22%)	1
No	381 (76.66%)	116 (23.34%)	
*Coeliac Positive*			
Yes	5 (83.33%)	1 (16.67%)	1
No	397 (76.64%)	121 (23.36%)	

^a^ Log transformed; ^b^ Poisson regression; ^c^ Multiple first-degree relatives with T1D; ^d^ Includes donor gametes with genetic first degree relative (FDR) with T1D.

## Data Availability

Deidentified individual participant data (including data dictionaries) may be made available, in addition to the study protocols, the statistical analysis plan, and the informed consent form. The data can be requested by researchers who provide a methodologically sound proposal that aligns with the goals of the ENDIA Study Group. Additional details can be found at: https://www.endia.org.au/for-researchers/ (accessed on 7 March 2023).

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
