# Peer review of "Experiences of Caregivers and At-Risk Children Enrolled in a Prospective Pregnancy-Birth Cohort Study into the Causes of Type 1 Diabetes: The ENDIA Study"

_children, 2023, doi:10.3390/children10040637_

Round 1

Reviewer 1 Report

Thank you for an interesting paper about the experience of caregivers and children with being involved in the Environmental Determinants of Islet Autoimmunity (ENDIA) study.

Generally, the paper is well-written, and I commend the authors for conducting a thorough study about participant experiences. Such studies are an underrated tool for improving participation rates and retain participants in the cohort. I think that studies aiming to understand the underlying reasons for drop-out are an important contribution to the research field and to future longitudinal cohort studies.

I further acknowledge the authors for including consumer representatives in their study.
I only have a few additional remarks that I would like the authors to consider:

The difference which are reported to be statistically significant (e.g. age of mothers and age of children who are completers and non-completers respectively). In continuation hereof I suggest that the authors discuss the clinical relevance of the identified differences.

In ’ Table 2. Demographics of caregivers with thoughts of leaving ENDIA’, would it make more sense to provide the percentages horizontally instead of vertically? E.g. Instead of ’Site A 129 (32.09%) 27 (22.13%)´ it would be: ’ Site A 129 (83%) 27 (17%). This way might be easier for the readers to see which site had the largest proportion of participants considering leaving.

In the manuscript, the discussion section includes traditional discussion points where the results are compared to other studies as well as potential changes to the study based on the results. These should be reported separately with the potential and planned changes reported in an ‘implications’ section.

Author Response

We wish to sincerely thank Reviewer 1 for their considered review and comments. To respond specifically to your points:

1. We have reflected on the clinical relevance of differences reported in table 1 (i.e. completer mothers were: older, children were younger, family more likely to have a father with T1D, were more active in the study, and attend particular study sites) and table 2 (those thinking of leaving had been in the study longer, children were older, had more days since previous visit and were participating at particular study site) in the discussion. Please see lines 258-263 and 285-287. We have addressed the implications of differences in satisfaction and response by site and this is, we feel, directly related to high staff changeover which is discussed from line 292.

2. We have made the alterations to table 2 as suggested; you will find this from line 209 onwards.

3. We have added an "implications" section as suggested; please see from line 315.

We hope these responses and edits meet your expectations. Thanks so much again

Reviewer 2 Report

McGorm et al. reported on an interesting study about experiences of caregivers and children enrolled in prospective study. Only minor changes are needed in the manuscript:

1. We advise to further expand the introduction about the ENDIA study in order to further clarify the aims and results of proscpective pregnancy birth cohort.

2. It is also interesting that almost 50% of respondents did not participate. In similar studies (TEDDY, ...) what was the response rate? And how do you comment on it.

Author Response

We wish to thank Reviewer 2 for their review and suggestions to improve the paper. In specific response to your points: 

  1. We appreciate the opportunity to expand on our prospective study and have added lines 61-70, including references to earlier results; we hope this provides greater context.
  2. Our response rate was comparable to the TEDDY study (50.5% v 59.1%) and this has been referred to in line 434. Reflections around the response rate are addressed in the limitations section from line 446. TEDDY did not report on their survey of children so we can not compare. 

Thanks again

Reviewer 3 Report

The manuscript of McGorm et. al "Experiences of caregivers and at-risk children enrolled in a prospective pregnancy-birth cohort study into the causes of type 1  diabetes: the ENDIA study" presents the results of the survey of satisfaction of caregivers and children participatin in ENDIA study. The results are important to understand the motivation and perception of burden by participants and thir caregivers. This can be useful for planning measures that may increase retention of participants in clinical trials and other types of prospective studies that require study site vists and performance of procedures, such as blood samples collection.  

Author Response

We thanks Reviewer 3 for their time to review this paper and provide their favourable comments.